# The Therapeutic Potential of Milk Extracellular Vesicles on Colorectal Cancer

**DOI:** 10.3390/ijms23126812

**Published:** 2022-06-18

**Authors:** Manal A. Babaker, Fadwa A. Aljoud, Faris Alkhilaiwi, Abdulrahman Algarni, Asif Ahmed, Mohammad Imran Khan, Islam M. Saadeldin, Faisal A. Alzahrani

**Affiliations:** 1Department of Biochemistry, Faculty of Science, King Abdulaziz University, Jeddah 21589, Saudi Arabia; m.babaker@mu.edu.sa; 2Department of Chemistry, Faculty of Science, Majmaah University, Al Majmaah 11952, Saudi Arabia; 3Regenerative Medicine Unit, King Fahd Medical Research Centre, King Abdulaziz University, Jeddah 21589, Saudi Arabia; faljoud@stu.kau.edu.sa (F.A.A.); faalkhilaiwi@kau.edu.sa (F.A.); 4Department of Natural Products and Alternative Medicine, Faculty of Pharmacy, King Abdulaziz University, Jeddah 21589, Saudi Arabia; 5Department of Medical Laboratory Technology, College of Applied Medical Sciences, Northern Border University, Arar 73221, Saudi Arabia; abdulrahman.eid@nbu.edu.sa; 6MirZyme Therapeutics, Innovation Birmingham Campus, Faraday Wharf, Birmingham B7 4BB, UK; asif.ahmed@mirzyme.com; 7School of Health Sciences, University of Southampton, University Road, Southampton SO17 1BJ, UK; 8Centre of Artificial Intelligence in Precision Medicines (CAIPM), King Abdulaziz University, Jeddah 21589, Saudi Arabia; mikhan@kau.edu.sa; 9Research Institute of Veterinary Medicine, Chungnam National University, Daejeon 34134, Korea; 10Laboratory of Theriogenology, College of Veterinary Medicine, Chungnam National University, Daejeon 34134, Korea; 11Embryonic Stem Cells Unit, Department of Biochemistry, Faculty of Science, King Fahd Medical Research Center, King Abdulaziz University, Jeddah 21589, Saudi Arabia

**Keywords:** colorectal cancer, milk exosomes, extracellular vesicles, characterization, therapeutic effects

## Abstract

Colorectal cancer remains one of the leading prevalent cancers in the world and is the fourth most common cause of death from cancer. Unfortunately, the currently utilized chemotherapies fail in selectively targeting cancer cells and cause harm to healthy cells, which results in profound side effects. Researchers are focused on developing anti-cancer targeted medications, which is essential to making them safer, more effective, and more selective and to maximizing their therapeutic benefits. Milk-derived extracellular vesicles (EVs) from camels and cows have attracted much attention as a natural substitute product that effectively suppresses a wide range of tumor cells. This review sheds light on the biogenesis, methods of isolation, characterization, and molecular composition of milk EVs as well as the therapeutic potentials of milk EVs on colorectal cancer.

## 1. Introduction

Colorectal cancer (CRC) has recently become increasingly malignant in the digestive tract, representing the third most common cancer in terms of incidence (10.2%) and mortality (9.2%) according to the WHO [1]. Despite significant advancements in chemotherapy, its severe disadvantages as well as the likelihood of therapy resistance and relapse due to colorectal stem cells (CSCs) diminish treatment efficiency [2]. Thus, CSCs are the cause of tumor initiation and their sustained growth [3].

Colorectal CSCs are chiefly generated from intestinal stem cells (ISCs) or differentiated intestinal cells that have acquired adequate genetic changes to cause tumor formation [4]. Furthermore, many similarities are present in the fundamental properties of colorectal CSCs and stem cells [5], such as the self-renewal ability and differentiation in several directions, excellent treatment resistance, and distant metastasis [6].

New therapeutic methods are required to destroy tumor cells and to avoid metastasis [7]. The possible beneficial impacts of the extracellular vesicles (EVs) of breast milk (BM) have recently gained a lot of interest in tumor treatment because they are less harmful and immunogenic than synthetic nanoparticles; being very small and their bi-layered lipid membrane nature permits them to traverse the blood–brain barrier (BBB) and cell membranes to deliver valuable genetic cargo that affects the surrounding and distant target cells [8,9,10,11,12,13,14,15].

Normal and abnormal cells, such as blood, amniotic fluid, bile, and milk, discharge EVs into the extracellular space [16,17]. For more than the last three decades, EVs have been believed to be cellular trash generated by cellular debris and to have no impact on adjacent cells [14]. However, their role in cell connection (through transmitting genetic material) and the immunological response has since been shown in previous research [18,19,20,21].

## 2. Extracellular Vesicles

Extracellular vesicles (EVs) are categorized into different forms according to their (a) size (e.g., small <200 nm, medium and/or large >200 nm); (b) biochemical composition (e.g., CD63+/CD81+-EVs or annexin V-stained EVs); and (c) cell of origin (e.g., apoptotic bodies, neuronal EVs, and podocyte EVs).

Several researchers used the terms exosomes (for vesicles with a range from 30 to 200 nm), microvesicles (ranging between 100 and 1000 nm), and apoptotic bodies (with a size of more than 1000 nm) according to the discharge mechanism of these EVs [22,23,24,25,26,27,28] (Figure 1). However, according to the regulations of guidelines that were published and circulated among the researchers (2018 Minimal Information for Studies of Extracellular Vesicles (MISEV)), authors are encouraged to use the generic term EVs with an operational term for the bilipid particles that are released from the cells, as mentioned above [29].

### 2.1. Exosome Biogenesis, Secretion, and Uptake

The biogenesis of exosomes consists of three steps. First, invagination of the cell membrane generates endocytic vesicles. Second, the endosomal membrane budding inward forms multivesicular bodies (MVBs), which include intraluminal vesicles (ILVs). Finally, MVBs merge with the cell membranes of different cell types, secreting ILVs as exosomes outside cells via exocytosis or breaking down MVBs through fusion to lysosomes [30,31].

Cells take up exosomes in many ways, including endocytosis [32], direct merger with the cell membrane [33], and receptor–ligand interactions [34] (Figure 2).

### 2.2. Techniques for Isolation of Milk EVs

Exosomes are now isolated and purified from cell cultures and bodily fluids using different methods [35]. However, the separated exosomes’ pureness varies, which is probably related to the contaminated particles, different sub-types of EVs, sample viscosity, and proteins of milk [36]. Furthermore, poor exosomal purity is because most existing separation methods fail to fully isolate exosomes from compounds that have identical biophysical characteristics, such as lipoproteins [35]. As a result, a combined enhanced protocol that was developed after a systematic evaluation of ultracentrifugation, ultrafiltration, poly-ethylene glycol-based precipitation, immunoaffinity capture, microfluidics, tangential flow filtration, and size-exclusion chromatographymethods to optimize exosome separation from several different body fluids, as the 2018 Minimal Information for Studies of Extracellular Vesicles (“MISEV”) guidelines reported [37,38], is recommended.

One of the most important challenges that researchers face during the isolation of milk EVs is the high content of protein (especially casein micelles) and lipoprotein. Therefore, milk samples should be processed for defatting and casein removal before starting EV isolation from the whey portion of the milk. Defatted milk can be performed through centrifugation at a lower force (1200× *g*, 4 °C, 10 min) to remove fat globules, cells, and cell debris [39]. Casein can be excluded by one of these three methods: (1) centrifugation at a mid-force (initially at 21,500× *g*, 4 °C, for 30 min and then repeated for 1 h); (2) the acetic acid precipitation method, which can be applied through the addition of acetic acid 17.5 *N* at a volume of 1:100 (acetic acid/milk) and centrifugation at a lower force (4500× *g*, 4 °C, 30 min); or (3) the ethylenediamine tetraacetic acid (EDTA) precipitation method, which can be performed through mixing 250 mM EDTA-3Na with the defatted milk for 15 min and then centrifuged at a lower force (4500× *g*, 4 °C, 30 min). The resulting defatted and de-caseinated portion should be filtered with 0.8 µm filters to result in a whey portion free of cell debris for EV isolation [39,40,41,42].

Milk, or any starting material type and its quantity; the accessibility of specialized devices; medical usage; and the desired outcomes are some of the essential factors for selecting the EV isolation approach [38]. Table 1 displays the advantages and drawbacks of each approach.

#### 2.2.1. Ultracentrifugation and Density Gradient Ultracentrifugation

Ultracentrifugation (UC) is classified as analytical or preparative. Particulate materials can be studied using analytical ultracentrifugation [43]. Another method is preparative ultracentrifugation, which is essential for exosome separation and for the separation of minute bioparticles [44]. Ultracentrifugation is considered the exosome separation gold standard and is commonly used and reported in procedures [43]. Moreover, ultracentrifugation is expected to be used in 56% of all exosome isolation procedures in exosome research [45]. Differential and density gradients are the two types of preparative ultracentrifugation [43]. Differential centrifugation separation utilizes a series of centrifugation operations that increase in velocity and duration [46]. The concept of this technique depends on large molecules separating first before smaller ones [46,47]. EVs can be isolated depending on their size, density, and mass, mostly by using density gradient ultracentrifugation using a sucrose gradient or an iodixanol gradient [43]. When compared with differential centrifugations, this method resulted in pure exosomes by isolating EVs from proteins and non-vesicular particles. The method depends on placing biological substances on top of the density gradient media and subsequent ultracentrifugation. Depending on the EV’s density, they appear as a separate layer, and then, for further purification, they are collected and ultracentrifuged [48]. However, the main drawbacks of this method are the possibility of losing the sample during separation and the complex process [49]. In addition, studies show that density gradient ultracentrifugation can be used to isolate EVs from bovine milk successfully [50].

#### 2.2.2. Ultrafiltration

Ultrafiltration (UF) uses membranes with certain pore sizes to separate the preset size range particles [51,52,53]. The primary concept of ultrafiltration is the separation of particles according to their volume and molecular weight using a filter membrane [38].

#### 2.2.3. Size-Exclusion Chromatography

The size-exclusion chromatography (SEC) method employs a biological fluid as a mobile phase and the stationary polymer of pored gel filtration [54,55]. Differential elution is possible due to the characteristics of the stationary phase: initially eluting larger and then smaller particles [38]. The main restriction of this method is the sample volume, which should be below 5% of the column volume. For this reason, SEC is not valid for the direct isolation of exosomes from bulk milk, and most researchers used it after other methods of isolation, especially ultrafiltration and ultracentrifugation, to obtain pure exosomes without protein contamination. In terms of the purity of milk EVs, combining the acetic acid precipitation and SEC method to yield high quantity of EVs with a high level of purity is recommended [39,42]. Additionally, sequential centrifugation followed by SEC was successfully used to isolate EVs from the milk of humans and cows [56].

#### 2.2.4. Polyethylene Glycol-Based Precipitation

Exosome precipitation, which anticipates polyethylene glycol (PEG) as a media, is a common way of isolating exosomes [57,58]. The idea behind this technique is that PEG bonds to water molecules, allowing exosome aggregates to form quicker, which can subsequently be precipitated using low-velocity centrifugation at 1500× *g* [53]. Exosome separation is also performed with PEG, which is commonly produced in kits such as ExoQuick [59].

#### 2.2.5. Immunoaffinity Capture

This approach works by separating certain exosomes, depending on their membrane protein expression. Antibodies are routinely used in certain exosome surface antigens, particularly the tetraspanins CD9, CD63, and CD81. By incubating specimens with magnetic beads [60] covered with antibodies against the antigens, exosomes can be isolated by immunoaffinity capture [61].

#### 2.2.6. Microfluidics

Exosome isolation using microfluidic instruments depends on many factors, such as immunoaffinity, density, and size [62]. In this technique, antibodies fixed on microfluidic equipment, commonly called chips, specifically bind to exosome antigens and isolate exosomes [49]. Moreover, microfluidic resistive pulse sensing (MRPS) has recently developed as a potent new method for detecting the size and concentration of EVs [63]. MRPS utilizes electrical sensing to determine the number and size of EVs directly and individually, without the use of any optics or mathematical algorithm. Therefore, MRPS does not depend on the material properties of EVs and can measure EVs precisely regardless of their polydispersity [63].

#### 2.2.7. Tangential Flow Filtration (TFF)

In tangential flow filtration (TFF), the fluid containing EVs is spread across instead of pushed through the filter, thereby constituting a pressure differential. The milk sample runs parallel to the filter and is reused several times across a reservoir [64]. TFF was successfully used to isolate highly pure EVs from cow milk on a large scale compared with the UC method [64]. Unlike direct filtration processes, TFF using two membranes with pore sizes of 200 and 30 nm connected to a peristaltic pump can overcome clogging problems during the isolation of milk EVs [65].

**Table 1 ijms-23-06812-t001:** Advantages and drawbacks of EV isolation approaches.

EV Isolation Approach	Advantages	Disadvantages
Ultracentrifugation (UC)	Simple to use, needs minimal technical experience, cost-effective (one ultracentrifuge machine for long-term usage), and requires little or no sample preparation [43]	Time-consuming, structural deterioration, and co-isolation of lipoproteins [28,54,66,67]
Ultrafiltration (UF)	Takes less time and effort [53] andgenerates very pure exosomes [68]	Employs power, which could result in a lack of exosomes due to membrane rupture and impurity of separated exosomes [69,70], andadherent particles also clog pores, resulting in a decrease in flow and elution performance [71]
Size-exclusion chromatography (SEC)	Fast, convenient, and inexpensive [54]	This method cannot distinguish between exosomes and similar-sized microvesicles [38]
Poly-ethylene glycol (PEG)-based precipitation	Handles multiple samples at once with convenience, speed, and relatively low costs without destroying the exosomes [58]	Other compounds such as protein can precipitate, contaminating the exosome [53,72]
Immunoaffinity capture	Shortens exosome separation periods and increases exosome purity [45,59]	Costly, ineffective, and not recommended for large-scale exosome separation [45,59]
Microfluidics	Effective and quick processing, and high pureness of exosome isolation [49]	Highly complicated and costly [49]
Tangential flow filtration	Fast and simple method [73]	The existence of nanoparticles of similar sizes to exosomes can be a limitation [73,74]

### 2.3. Techniques for Characterization of EVs

The main focus of the Minimal Information for Studies of Extracellular Vesicles (MISEV) 2018 is EV characterization development, using markers such as lipid as well as proteins, which tend to be very helpful in demonstrating the EVs’ general structure. The markers used to characterize EVs can vary, depending on the cells of origin [29]. Nanoparticle tracking analysis, transmission electron microscopy (TEM), dynamic light scattering, flow cytometry, and Western blot are methods that have been used for exosomal characterization [35]. Two types of techniques can be used for EV characterization. First, the physical characterization technique discovers the shape and molecular dimensions. Second, biochemical characterization identifies the protein of the membrane and the lipid composition [73]. Table 2 describes the advantages and drawbacks of exosome characterization techniques.

Generally, EVs can be visualized with TEM, and a more detailed resolution can be attained by cryo-TEM [47]. EVs are lipid bilayer membranous vesicles with heterogeneous sizes (i.e., the average size of milk EVs ranges from 160 to 190 nm based on their method of isolation [39]). As examined by TEM, camel milk exosomes isolated by differential ultracentrifugation appeared spherical, with diameters ranging from 30 to 100 nm [7,75,76]. Human milk exosomes have similar shapes and sizes [77]. Similar shapes were observed in other animal milk exosomes isolated by ultracentrifugation but with different sizes (80–130 nm) in cow [78], (30–200 nm) [79] in buffalo, (70–170 nm) in goat [80], and (50–100 nm) in pig [81]. DLS revealed nearly similar size distributions of milk exosomes in yaks (131.1 ± 53.25 nm) and cows (131.5 ± 52.39 nm) [82].

As the sources of exosomes are variable, many proteins can be found within the exosomes or at their surfaces. Exosomal proteins including MHC I, MHC II, and heat shock proteins as well as specific surface exosomal markers (such as CD9, CD63, and CD81) can be detected and confirmed using Western blotting or flowcytometry [83,84,85]. Comparedwith other EVs, exosomes lack integrin-β1, p-selectin, CD40, and calnexin but contain some proteins associated with exosome biogenesis such as Alix and Tsg101 [86].

#### 2.3.1. Dynamic Light Scattering (DLS)

Dynamic light scattering (DLS) uses a laser beam transmitted over a solution of nanoparticles [87]. When the laser light hits EVs, the light scatters in different directions. By calculating the intensity of the scattered light per time unit, its oscillations can be noticed as a result ofthe Brownian motion of suspended EVs. The main advantage of this method is its capability to quantify a wide range of particles (from 1 nm to 6 μm). However, this method could not determine the accurate size of particles originating from different sources or with variable size (polydispersed suspensions). Consequently, the presence of larger particles in the sample, even at a low concentration, masks the recognition of smaller particles, causing false results [88]. Therefore, to obtain more accurate results, we should remove any large contaminates. DLS is the most preferable technique for determining the size, distribution, mobility, surface charge, and concentration of exosomes [87]. DLS was also used to determine the distribution and size of EVs isolated from blood [89] and ovarian cancer cells [90]. In general, the DLS method can determine the diameter range of EVs, but cannot determine the source of these EVs [90].

#### 2.3.2. Nanoparticle Tracking Analysis (NTA)

Nanoparticle tracking analysis (NTA) depends on Brownian movements, which explain the random variations of molecules in a liquid solution. The results can be analyzed using the NanoSight device. The apparatus brightens individual nanoparticles with a laser beam, whereas a camera follows and registers their Brownian motions by measuring the dispersed illumination [91]. Similar to DLS, NTA also can determine particle size and distribution [87,92].

#### 2.3.3. Transmission Electron Microscopy (TEM) and Cryo-TEM

The concept of transmission electron microscopy (TEM) depends on the generation of pictures by a stream passing electrons via a specimen, where a secondary electron can be formed. Specific lenses can be used to gather and magnify electrons [93]. Cryo-TEM shows high-resolution EVs and provides potentially a more realistic morphology of EVs [94].

#### 2.3.4. Western Blot

Exosomal proteins and particular surface indicators, such as MHC I, MHC II, tetraspanins, and heat shock proteins, can be detected and confirmed using Western blotting (WB) [83,84,85]. This approach is also called immunoblotting because antibodies are employed to identify unique proteins in complicated protein specimens. The method combines many systems and involves separating proteins on a gel, transferring the proteins to a hard surface, and identifying the proteins of significance [95].

**Table 2 ijms-23-06812-t002:** EV characterization techniques’ advantages and drawbacks.

EV Characterization Techniques	Aims	Advantages	Drawbacks
Dynamic light scattering (DLS)	Identifying EV distribution and size [87]	The best method for measuring a single type of nanoparticle in a solution and is easy to be performed [96]	Does not allow for the nanoparticles to be visualized [96]
Nanoparticle tracking analysis (NTA)	Identifying EV distribution and size [91]	Determination of very small EVs up to 30 nm, sample preparation does not affect the morphology of EVs, very quick and easy sample preparation, and samples can be reused after the measurements and can detect fluorescently labeled antibodies targeting EVs antigens [87,92]	Masking of small size particle by large size particles, as in DLS, finding the most appropriate dilution factor to obtain resonant results, and the fluorescent signal should be very bright to detect the EV phenotype accurately. Therefore, it is recommended to use antibodies coupled with quantum dots (Q-dots), which are very bright fluorochromes [87,92]
Transmission electron microscopy (TEM)	Identifying EV form [93]	The interior morphology of a particle may be seen via TEM, which can also reveal details on the size of the particle [97]	Complex sample preparation (numerous processes and EV morphological alteration). Biological specimens can be destroyed [93].
Western blot (WB)	Identifying EV marker protein expression [83,84,85]	Evaluates marker proteins in both qualitative and quantitative ways [97]	Complicated and consumes an extended period [29].
Flow cytometry	Identifying EV biomarkers [87]	Provides high-speed analysis and needs minimal sample concentration [97]	Takes a lot of time and is very complicated [97]

#### 2.3.5. Flow Cytometry

This approach relies on a laser bar with a chosen frequency that is coordinated through a liquid flood containing suspended molecules. Light-level dispersing relies on the particles present inside the samples. Furthermore, this method estimates particles marked with fluorescent pigments. If this is upheld, flow cytometry can investigate a particle’s overall dimensions and granulation [87].

### 2.4. Bioactive Compounds of Milk EVs

Human colostrum and breast milk exosomes were first separated and characterized in 2007. In addition, Artiodactyla mammals’ milk exosomes were separated, and portrayals were examined and reported [7,79,98,99,100]. Exosomes from milk may be discharged by mammary gland epithelial cells. Furthermore, during breastfeeding, exosomes are delivered from milk fat globules [101,102]. Munagala et al. provided a detailed procedure for milk exosome separation and characterization [103]. As shown in Figure 3, proteins, lipids, and nucleic acids are milk exosomes’ organic elements.

#### 2.4.1. Proteins of Milk EVs

Unmistakably, the physiology of milk EVs relies heavily on proteins [28,103,104]. The milk EVs’ protein percentage changes depending on the host’s age, lactation stage, activity level, host illnesses, and diet [105]. Moreover, the EVS separation process and EV sources influence the amounts and variations of EV proteins [28,48,102]. EVs have many types of proteins [106]. Around 1963, 2107, and 639 proteins were detected in human [107], bovine [108], and pig [109] milk EVs, respectively. We summarize the top proteins that has been confirmed by Western blotting in milk EVs in Table 3. Most of these proteins are involved in the regulation of inflammation and cell proliferation, suggesting that milk EVs may affect the infant’s immune system and gastrointestinal development [107]. However, proteomics analyses of milk EVs and confirmation through the Western blot technique are still in their infancy, and the mechanism by which EV-derived proteins can exert their biological functions has not yet been elucidated. Rab proteins are tiny GTPases that are part of the Ras superfamily. They have a critical function in regulating vesicle budding, motility, and fusion [20,110]. Moreover, Alix (programmed cell death 6 interacting protein PDCD6IP), TSG101 (tumor susceptibility gene 101), other endosomal sorting complex proteins [111], and proteins involved in miRNA binding and transferring target cell identification, and merging may be found in all exosomes (tetraspanins CD9, CD63, and CD81). Tetraspanins are important structural elements of exosomal membranes that promote exosome attachment on the surface of the target cells and are required for exosome production as well as merger activities [20,112,113,114]. Exosomes may also include a variety of enzymes, including proteases, peroxidases, lipid kinases, and some catalytic proteins [115]. Exosomes are full of cytoskeleton proteins (actin, tubulin, and cofilin) and heat shock (HSP60, HSP70, and HSP90) [116,117]. Integrins in exosomes derived from milk are critical indicators of EVs’ internalization and bioactivity and serve as delivery direction predictors [118]. All types of exosomes, including milk exosomes, contain all of the proteins mentioned above, while milk exosomes contain unique milk proteins, such as caseins, lactoglobulin, lactoferrin, CD36, and the polymeric immunoglobulin receptor forerunner [119]. Moreover, milk exosomes are indicated by butyrophilin, lactadhedrin, and xanthine dehydrogenase [106,120,121]. Furthermore, markers, such as integrin-β1, p-selectin, CD40, and the endoplasmic reticulum (ER) marker calnexin, cannot be found on exosomes surfaces since they are counted as other multivesicular body markers [28,86].

#### 2.4.2. Lipids of Milk EVs

Exosome membranes are full of different types of lipids, including phosphatidylcholine, cholesterol, sphingomyelin, and ceramides [126]. Various kinds of lipids are circulated unevenly in exosomes membranes. Thus, sphingolipids and glycosphingolipids such as gangliosides, can be situated in the external membranes, while various kinds of lipids are situated in the inner membranes [106,127]. The lipids of the exosome membranes are involved in vesicles’ biogenesis and influence their bioactivity; they are not inert molecules [128].

#### 2.4.3. Nucleic Acid of Milk EVs

Investigations during the last 20 years have found different noncoding RNAs [129]: in bovine [101], human [130], panda [131], porcine [18], and rodent [132] milk. The exosomal RNA’s benefits include steadiness in the presence of RNases, and a low intestinal PH [40]. Exosomes also include messenger RNA (mRNA) and thousands of microRNA (miRNA), which can be transmitted into the target cell and may transport new genetic information [133]. The target cell’s protein expression may be altered by transmitting new genetic information. As a result, they may take part in protein expression and signaling cascades between cells [75,134]. In addition, these elements play an important part in the immune system’s development, inflammatory regulation, and cell proliferation and progression [36,48]. The exosomal mRNA of mice can be translated into proteins in human cells, for example, when human cells are treated with mouse exosomes [135]. Milk EVs contain overexpressed miRNAs, suggesting a conserved release of specific milk miRNAs that are mostly linked to cellular defense mechanisms, and anti-inflammatory and immunomodulatory potential [136,137,138]. For instance, among the top miRNAs found in human and cow milk are miR-30d-5p, miR-148a-3p, miR-200a-3p, miR-200c-3p, let-7a-5p, and let-7f-5p (Table 4). Moreover, miR-22-3p (stem cell differentiation and inflammatory prevention) and miR-146a (prevention from hypoxic damage in intestinal epithelium) were also detected in milk EVs [52]. Interestingly, miR-148a-3p works as a regulator of the DNA methyl-transferase 1, which raises concerns about the impact of recurrent milk consumption on epigenetic regulation of the human genome [139]. As a consequence, milk exosomes seem to be potential candidates for creating novel therapeutic methods for a variety of illnesses, particularly cancer [121,140].

### 2.5. Therapeutic Potential of Milk EVs and Cancer

According to Munagala et al., cow milk-derived exosomes were reported to have an inherent anti-cancer potential by reducing the growth of malignancies, such as colon and ovarian tumors. As surveyed by the MTT test, cancer cells treated for 72 h using 50 μg/mL of exosomal proteins decrease growth by 8–47%, recommending using the exosome as an anti-tumor drug carrier [146]. Furthermore, camel milk and its contents have been shown to have anti-tumor impacts on the hepatoma cell line (HepG2), breast cancer cell line (MCF7), and mouse hepatoma cell line (Hepa 1c1c7) [147,148]. Camel milk suppresses cell development and causes apoptosis in HepG2 and MCF7 cells by stimulating caspase-3, in addition to the death receptor DR4, and accumulating superoxide radicals inside the cells [146]. Furthermore, after studying camel milk and exosomes derived from camel milk, it was found that they suppress MCF7 cell proliferation, which is accompanied by a decline in MCF7 cell migration [7]. Rats that had cancer showed significant improvements after the use of milk of camel milk and exosomes derived from it. Camel milk as well as its derived exosomes fundamentally diminish cancer weight, stop tumor development, and improve the immune system. While exosomes have a greater anti-cancer impact in general, the number of splenic T lymphocytes in rats given camel milk increased significantly, demonstrating that there are more immune-stimulating elements in camel milk compared with exosomes [7]. According to El-Kattawy et al., exosomes derived from camel milk had a specific antiproliferative impact on tumor HepaRG cells but no toxic impact on regular liver THLE-2 cells. The anti-tumor impact may well be related to the stimulation of apoptosis, as well as the prevention of inflammatory and angiogenesis. The findings suggest that exosomes obtained from colostrum are more effective at inhibiting tumor growth in HepaRG cells compared with exosomes isolated from the other lactation periods [149].

Additionally, a combination of exosomes derived from camel milk, hesperidin, and tamoxifen exhibited anti-tumor actions in MCF7 xenografts in mice and against MCF7 cells by inducing apoptosis and inhibiting invasion, migration, and angiogenesis.

Combining tamoxifen, hesperidin, and camel milk exosomes reduced the unfavorable effects of tamoxifen. This shows that hesperidin and camel milk exosomes may have significance in the treatment of breast cancer as additives to tamoxifen [150].

### 2.6. Therapeutic Potential of Milk EVs in Colorectal Cancer

Consuming fermented milk products does not have a preventive effect on the progress of colorectal cancer compared to patients consuming raw unfermented milk [151]. A systematic review showed that a daily increment of 200 g of milk consumption could reduce colorectal cancer danger [152]. Milk exosomes are negatively affected by the fermentation process. Indeed, cow milk exosomes’ size and protein content were mainly diminished in fermented cow milk, with severe losses in miRNA-29b and miRNA-21 [153,154]. Breast milk-derived exosomes can selectively promote normal colon epithelial cell proliferation but with no effect on colonic malignancy cells [155]. On the other hand, cow milk exosomes have potent direct anti-tumor effects against colorectal cancer [146]. In contrast, another study reported that the incubation of cow milk exosomes with CaCo-2 cells maintained their metabolic activity and improved cell survival but did not trigger cell proliferation [122].

Chronic inflammation of the intestine mediated by cytokines such as TNFα, TGF-β, and IL-6 can be a common element in colorectal cancer development [156]. TGF-β and miR-155, two milk exosome components, inhibited T lymphocytes in the intestine, thereby suppressing colitis progress [157]. MiRNA-148a can significantly affect immune control and cancer development [158]. MiR-148a reduces the production of cytokines, such as TNFα, IL-6, and IL-12, in addition to the innate response and antigen presentation of Toll-like receptor (TLR)-stimulated dendritic cells when it targets calcium/calmodulin-dependent protein kinase II (CaMKII) [159]. MiR-148a expression is decreased in colorectal cancer cells [160,161,162,163], and this downregulated expression activates DNA-methyltransferase 1 (DNMT1) [164,165,166]. The incubation of colorectal cancer cells (Lim 1215) with exosomes derived from human milk elevated the level of miR-148a in the cells but decreased DNMT in the normal colon epithelial cell line (CRL 1831) [167]. Milk exosomal miR-148a targets DNMT1 and therefore inhibits the activity of this important activator of colorectal cancer [168]. A higher level of Rho-associated coiled coil-containing protein kinase 1 (ROCK1), which is considered a major miR-148a target, plays a crucial role in the development of colorectal cancer [169,170]. Therefore, the carcinogenesis of colorectal cancer may be affected by the transport of miR-148a in milk exosomes [171].

### 2.7. The Use of Milk EVS for Drug Delivery in Colorectal Cancer

The three main advantages of EV nanocarriers in drug delivery are as follows: (1) The phospholipid bilayer of the EV membrane shields the content of the EVs from destruction [172,173,174,175]. (2) EVs include membrane proteins (CD9, CD63, CD81, and others) and membrane-associated proteins on their surfaces, which may significantly extend the duration of exosome circulation in the blood and improve drug delivery to specific tissues [48,55,176,177,178]. (3) EVs can cross physiological boundaries, such as the blood–brain barrier, blood–testis barrier, and cell membrane [173,179,180,181,182]. Currently, to transport targeted drugs, EVs derived from cancer cell lines, lymphocytes, and stem cells are used [176,183,184]. On the other hand, cell lines generate a small number of EVs, which makes it impossible to attain the quantities necessary for industrial pharmaceutical manufacture [185]. In addition, when delivered systemically, exosomes’ protein components may elicit immunological responses. In comparison, bovine milk EVs can be obtained in scalable quantities, as shown in some studies [176,186]; simultaneously, milk EVs do not yield systemic toxic effects or anaphylaxis in animal models [187].

Previous experiments on Caco-2 cell lines have shown that curcumin encapsulated in milk exosomes may cross the gastrointestinal barrier into the circulatory system and provide an increased antiproliferative impact [188]. As a result, curcumin, which may be used as a possible anti-cancer drug, is delivered by milk exosomes because it considerably enhances the stability, solubility, and bioavailability of adverse conditions in the gastrointestinal tract compared to free curcumin [3,189].

Additional research found that exosomal formulations of anthocyanidin (ExoAnthos) enhanced the stability of and anti-cancer activities in a variety of tumors, including HCT116 human colorectal cell lines. Anthos encapsulated onto exosomes of milk could increase medicinal effectiveness while avoiding hazardous adverse effects. Therefore, exosomes offer a safe and efficacious replacement for the oral administration of Anthos to cure various tumors [145].

Previous studies have shown that encapsulating siRNA in exosomes derived from milk resists severe digestive systems, optimizes intestinal permeability, and protects payloads on Caco-2 cells [190]. In addition, milk exosomes may be used to deliver siRNAs. After siRNAs are loaded to exosomes derived from milk using electroporation and chemical transfection, their activities of gene silencing were examined in vitro in a variety of tumors [191]. The uptake of exosomes that have siRNA in tumor cells causes the target genes to be silenced and to resist RNase [188].

Furthermore, exosomes coming from cow milk are effective miRNA carriers [191]. The target gene delivery performance was explained by examining the absorbance of the miR148a-3p-loaded cow milk exosomes in the Caco-2 cell lines. A gene microarray analysis revealed that cow milk exosomes could be employed as nanocarriers of efficient miRNAs that could develop future miRNA-based gene treatments [192]. EVs have gained attention as potential drug delivery vehicles due to their potential safety profile. Furthermore, milk EVs have also shown the potential to become drug delivery vehicles as EVs are less likely to evoke an immune response [193].

## 3. Conclusions

EVs are abundant in milk, as they are in other bodily fluids. These membranous nanoparticles have a critical function in intercellular connections, and they can be superior nanocarriers for proteins, messenger RNAs, and miRNA. Because of their potential anti-cancer properties as well as their non-toxic and non-immunogenic features, milk EVs have gained considerable interest. In vitro and in vivo, milk EVs influence immunological function and inhibit the growth of certain tumor cells. Additionally, since milk EVs express a high level of miRNA-148a, they may be employed to compensate for the miRNA-148a deficit in colorectal cancers, inhibiting the development of colorectal cancer. Furthermore, milk EVs may potentially be employed to treat colorectal cancer by acting as carriers of natural components and nucleic acids. Given the undeniable advantages of camel and cow milk EVs, their potential applications in cancer therapies are unlimited. Establishing how the different active compounds of camel and cow milk EVs perform biological roles, particularly in colorectal cancer treatment, is necessary for future research. Moreover, any potential adverse effects of milk EV therapy should be highlighted. Furthermore, it would be critical to confirm the EVs’ quality before clinical use by developing a standardized procedure for isolating, purifying, and manipulating milk EVs. Finally, as with other species, a camel’s dietary habit might impact the nutritional and EV contents. This impact will need to be further investigated in the future.

## Figures and Tables

**Figure 1 ijms-23-06812-f001:**
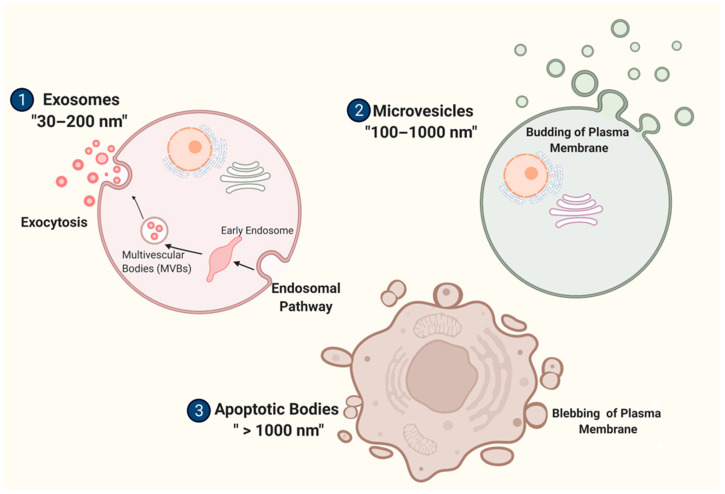
Classification of extracellular vesicles (EVs) with different discharge mechanisms. (**1**) Exosomes are made by endocytosis pathway, and discharged via exocytosis at a diameter of “30–200 nm”. (**2**) Plasma membrane budding forms the microvesicles (MVs), and they vary in diameter from 100 to 1000 nm. (**3**) With a size above 1000 nm, apoptotic bodies are discharged from the cell membrane by blebbing processes. Created with BioRender.com (accessed on 3 January 2022).

**Figure 2 ijms-23-06812-f002:**
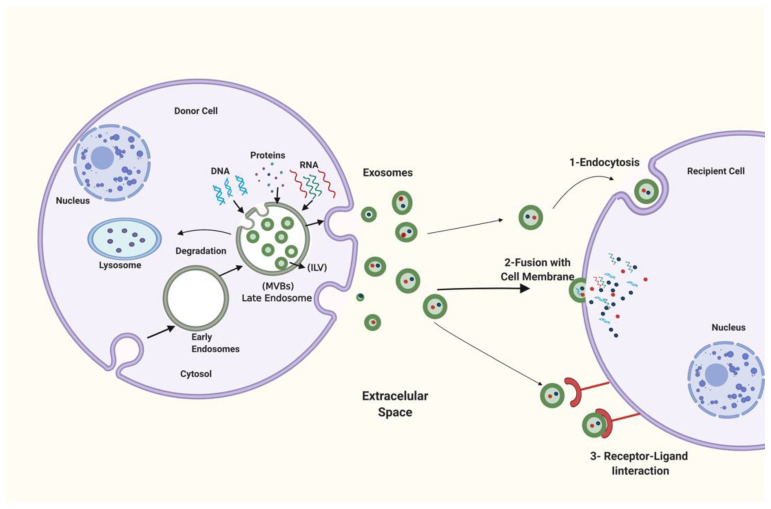
Exosome formation, discharge, and uptake. Late endosomes, often known as Multivesicular bodies (MVBs), form exosomes. Multivesicular bodies’ membranes bud inward and generate various sizes of exosomes known as intraluminal vesicles (ILVs). In this step, RNAs, proteins, and DNA are loaded onto exosomes. Multivesicular bodies’ may be degraded in the lysosome or released through the extracellular fluid by merging into the cell membrane. After that, the three ways that exosomes engage with the receptor cell are (**1**) endocytosis, (**2**) a direct fuse between the exosome membrane and the cell membrane, and (**3**) an interaction between the ligand and receptors. Created with BioRender.com (accessed on 23 February 2022).

**Figure 3 ijms-23-06812-f003:**
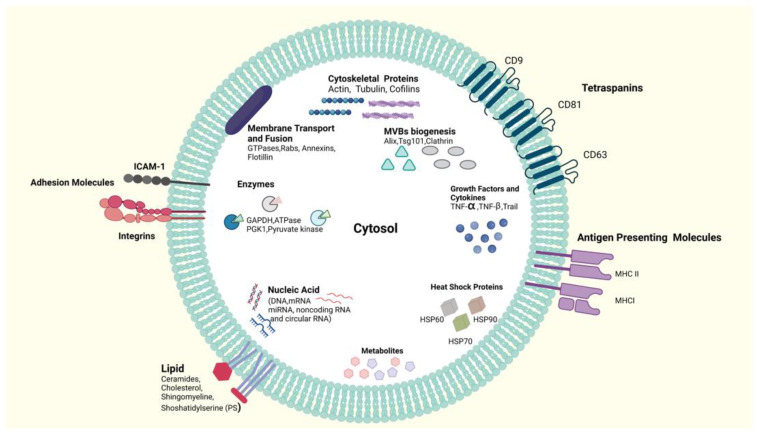
The main components of an EV. EVs are packed with a diverse array of molecules, such as lipids, proteins, nucleic acids (DNA, mRNA, miRNA, non-coding RNA, and circular RNA), and metabolites. In addition to sphingomyelin, phosphatidylserine (PS), cholesterol, and ceramides make up the lipid bilayer of EVs. EVs include tetraspanins, antigen-presenting molecules, and adhesion molecules. Furthermore, heat shock proteins (HSPs), cytoskeletal proteins, MVB biogenesis, enzymes, membrane transport, fusion proteins, growth factors, and cytokines are all proteins present in the EV lumen. Abbreviations in the figure: MVBs, multivesicular bodies; GAPDH, glyceraldehyde3-phosphate dehydrogenase; HSP, heat shock protein; MHCI, major histocompatibility complex class I; MHCII, major histocompatibility complex class II; miRNA, microRNAs; Tsg101, tumor susceptibility gene 101; TNF, tumor necrosis factor; TRAIL, TNF-related apoptosis-inducing ligand; ICAM-1, intercellular adhesion molecule 1; and PGK, phosphoglycerate kinase 1.

**Table 3 ijms-23-06812-t003:** Top proteins identified in milk EVs that were identified by the Western blot technique.

Species	Protein	Functions	References
Human	Oleoyl-ACP Hydrolase	Metabolism	[107]
	Parathyroid Hormone-Related Protein	Endocrine Functions and Epithelial–Mesenchymal Interactions	[107]
	Myelin Protein Zero-Like Protein 1	Immunoglobulin Superfamily and a Receptor of Concanavalin A	[107]
	EH Domain-Containing Protein 3	Cholesterol and Sphingolipid Transport	[107]
	Heat Shock Cognate 70	Protein Homeostasis in Stressed and Non-Stressed Cells	[122]
	Heat Shock Protein 70		
Cow	Butyrophilin, Xanthine Oxidase, Adipophilin, and Lactadherin	Milk Fat Globule Membrane (MFGM) Proteins	[108]
	MHC Class I	Immune Response	
Pig	EGF, TGFβ-3, MSTN, CTGF, IGFBP-7, PDGFA, HTRA3, THBS1, and Lactoferrin	Acute Inflammatory Response, Complement Activation, Classical Pathway, B Cell-Mediated Immunity, Negative Regulation of Blood Coagulation, Activation of Immune Response, and Protein Maturation and Processing	[109]
Camel, cow, human, and pig	Tumor Susceptibility Gene 101 Protein (TSG101)	Vesicle trafficking	[75,107,108,123,124,125]

**Table 4 ijms-23-06812-t004:** Top miRNAs identified in milk EVs from humans and cows.

Species	miRNAs	References
Human	miR-30d-5p, miR-148a-3p, miR-200a-3p, miR-200c-3p, let-7a-5p, miR-200b-3p, miR-21-5p, let-7b-5p, hsa, let-7f-5p, miR-30a-5p.	[138]
	miR-148a-3p, miR-30b-5p, let-7f-5p, miR-146b-5p, miR-29a, let-7a-5p, miR-141-3p, miR-182-5p, miR-200a-3p, miR-378-3p.	[141]
	miR-148a-3p, miR-22-3p, miR-30d-5p, let-7b-5p, miR-200a-3p, let-7a-5p, let-7f-5p, miR-146b-5p, miR-24-3p, miR-21-5p	[142]
	miR-22-3p, miR-148a-3p, miR-181a-1, miR-30d-5p, miR-141-3p, miR-26a-5p, miR-30b-5p, miR-92a-3p, miR-375-3p, miR-182-5p	[143]
Cow	mir-148a-3p, let-7a, let-7b, miR-21-5p, miR-99a-5p, let-7f-5p, let-7c, mir-200c, miR-26a-5p, miR-30d-5p	[144,145]

## Data Availability

Not applicable.

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
