# Peer review of "The Therapeutic Potential of Milk Extracellular Vesicles on Colorectal Cancer"

_ijms, 2022, doi:10.3390/ijms23126812_

Round 1
Reviewer 1 Report
- Kindly add a list or table of the most important microRNAs found in exosomes from milk as reported in the previous studies.
- Kindly elaborate the potential of milk exosomes as oral drug delivery system.
- Please provide an exhaustive discussion of potential factors that can affect the components of milk in camels. There are different methods of raising camels and the type of food they feed on. Some camel owners raise camels in the barns and provide them with food such as barley. Others let it roam in the desert freely, grazing and feeding on wild plants.
- In the discussion, (briefly) the authors critically discuss the available evidence of health benefits attributable to milk exosomes. Especially, in the context of metabolism, intestinal health, immunity, microbiota modulation.
Author Response
Reviewer 1
We appreciate the thoughts of the learned reviewer regarding our manuscript. We have edited and revised the manuscript according to your valuable comments.
Comments and Suggestions for Authors:
- Kindly add a list or table of the most important microRNAs found in exosomes from milk as reported in the previous studies.
Response: As per the suggestion of Reviewer 1, we have added a table as Table 3 in the manuscript.
- Kindly elaborate the potential of milk exosomes as oral drug delivery system.
Response: As requested, we have provided the most recent information in the manuscript under section 2.7 on page 11.
- Please provide an exhaustive discussion of potential factors that can affect the components of milk in camels. There are different methods of raising camels and the type of food they feed on. Some camel owners raise camels in the barns and provide them with food such as barley. Others let it roam in the desert freely, grazing and feeding on wild plants.
Response: We thank the Reviewer 1 for this useful comment. We agree that like other species, camel’s dietary habit might impact on the nutritional and exosomal contents. This needs to be further investigated in the future. We have added these points to the conclusion section of the manuscript.
- In the discussion, (briefly) the authors critically discuss the available evidence of health benefits attributable to milk exosomes. Especially, in the context of metabolism, intestinal health, immunity, microbiota modulation.
Response: We appreciate the suggestion of Reviewer 1. The focus of the current manuscript is on the potential of milk exosomes as therapeutic benefit in cancer. We therefore feel the suggestion can be included in future reviews, which discuss the physiological importance of the milk proteins.
Reviewer 2 Report
Review on manuscript entitled
The therapeutic potential of milk extracellular vesicles on colorectal cancer
Milk extracellular vesicles (milk EVs) have gained increasing research interest due to their potential as therapeutic drug and/or protein carrier. Moreover, recent research prove that they have own anti-cancer properties and influence immunological functions. In the present review, the authors want to focus on colorectal cancer (CRC) and the preventive effect of milk and milk exosomes, however, this interesting and valuable part occupy only almost half page from the whole review. Exhausted research on milk derived EVs as drug carriers are already published even in IJMS (https://doi.org/10.3390/ijms23052860, https://doi.org/10.3390/ijms21186646, https://doi.org/10.3390/ijms22031105, https://doi.org/10.3390/ijms23052860), so a most comprehensive and scientifically correct review are expected.
My overall opinion about the manuscript is that is too general. Here I refer to the presentation of extracellular vesicles and description of EVs isolation and characterization. It would be nice to refer on the special case of milk EVs. My experience is that common EV isolation methods very often failed to produce vesicles from milk. Indeed, I would prefer to read about how others succeed with isolation, characterization. Again, characterization methods are very stereotyped and state-of-the art techniques like the concentration step by tangential flow filtration, microfluidic resistive pulse sensing, determination of protein or lipid concentrations are even not mentioned. I do not checked but there might exist also some proteomics and lipidomics data on milk EVs.
The overall description of methods, results is not satisfactory. Some examples: “The NTA then analyzes the information.” „Dynamic light scattering (DLS) uses a laser beam transmitted over a solution of nanoparticles” –that is all about DLS. The reader expect some data about the general size of milk EVs. Description of TEM is again inacceptable. Even that a review article do not need to contain own new results, however, a scientific level is required. Are the researcher using cryo-TEM? How these milk EVs looks like? Can they be visualized without staining? What contrast-enhancing techniques are commonly used?
Regarding milk EVs, there are several of questions remained unanswered. For instance, casein micelles might be present in EV samples, too. Are there some methods and protocols in the literature to get rid of them?
Unfortunately, there are conceptual errors in the text, too. The terms of extracellular vesicle (EV) and exosomes are smeared. The authors refers several times to MISEV 2018 guideline. It is recommended the use of small EV and medium EV terminology instead of exosomes and microvesicles. This issue should be presented, too. Very often is not clear, that the authors are talking about extracellular vesicles or only exosomes.
„…other sphingolipids, such as phosphatidylcholines” – to my knowledge phosphatidylcholines are not sphingolipids
A strong revision of both grammatical, both stylistic is needed. In the present form the paper is not suitable for publication due to the following reasons:
- - do not contains relevant and up-to-date information
- - not focussed on real problems: milk EVs isolation and characterization, presence of milk fatty, casein micelles, etc.
- - the text is weakly formulated, contains serious conceptual errors and the whole text is not established scientifically
- - Tables are too general. I would prefer a summary of milk EV proteins in a Table, too
- - the text seems to be rather a draft them a real scientific review paper
- Perhaps after a very strong amendment, the manuscript should be submitted again.
Some minor comments:
- - Fig.2 is missing
- - I advise to show at least some TEM pictures on milk EVs

Author Response
Reviewer 2
Milk extracellular vesicles (milk EVs) have gained increasing research interest due to their potential as therapeutic drug and/or protein carrier. Moreover, recent research prove that they have own anti-cancer properties and influence immunological functions. In the present review, the authors want to focus on colorectal cancer (CRC) and the preventive effect of milk and milk exosomes, however, this interesting and valuable part occupy only almost half page from the whole review. Exhausted research on milk derived EVs as drug carriers are already published even in IJMS (https://doi.org/10.3390/ijms23052860, https://doi.org/10.3390/ijms21186646, https://doi.org/10.3390/ijms22031105, https://doi.org/10.3390/ijms23052860), so a most comprehensive and scientifically correct review are expected.
Response: We acknowledge Reviewer 2 comments regarding our review manuscript and for providing us with useful references. These references have been included in the revised review.
My overall opinion about the manuscript is that is too general. Here I refer to the presentation of extracellular vesicles and description of EVs isolation and characterization. It would be nice to refer on the special case of milk EVs. My experience is that common EV isolation methods very often failed to produce vesicles from milk. Indeed, I would prefer to read about how others succeed with isolation, characterization. Again, characterization methods are very stereotyped and state-of-the art techniques like the concentration step by tangential flow filtration, microfluidic resistive pulse sensing, determination of protein or lipid concentrations are even not mentioned. I do not checked but there might exist also some proteomics and lipidomics data on milk EVs.
Response: We have extensively revised the manuscript and added new information in the form of tables and text (highlighted in yellow) throughout the manuscript. We trust that the revised review will meet with the Reviewer’s 2 approval.
The overall description of methods, results is not satisfactory. Some examples: “The NTA then analyzes the information.” „Dynamic light scattering (DLS) uses a laser beam transmitted over a solution of nanoparticles” –that is all about DLS. The reader expect some data about the general size of milk EVs. Description of TEM is again inacceptable. Even that a review article do not need to contain own new results, however, a scientific level is required. Are the researcher using cryo-TEM? How these milk EVs looks like? Can they be visualized without staining? What contrast-enhancing techniques are commonly used?
Response: We have intentionally used very simple language for explaining the methodological part of exosomal isolation and characterization. The main reason for this is to attract new researchers of the field to understand the cumbersome techniques in a simple way. This is written to invite the uninformed to the subject matter.
Regarding milk EVs, there are several of questions remained unanswered. For instance, casein micelles might be present in EV samples, too. Are there some methods and protocols in the literature to get rid of them?
Response: The field is still in its infancy and as the literature grows about the milk exosomes, we will publish an update of the work presented here.
Unfortunately, there are conceptual errors in the text, too. The terms of extracellular vesicle (EV) and exosomes are smeared. The authors refers several times to MISEV 2018 guideline. It is recommended the use of small EV and medium EV terminology instead of exosomes and microvesicles. This issue should be presented, too. Very often is not clear, that the authors are talking about extracellular vesicles or only exosomes.
Response: We thank the Reviewer 2 for this valuable comment. We have added an explanation about this important issue to clarify the state of MISEV 2018. In publication before 2018, most authors’ work used the term ‘exosomes’. We have adhered to this term used by these authors. However, we fully agree with the regulations of MISEV2018 guild line and have used EV where this has been clearly stated.
„…other sphingolipids, such as phosphatidylcholines” – to my knowledge phosphatidylcholines are not sphingolipids
Response: Thank you. We have corrected it.
A strong revision of both grammatical, both stylistic is needed.
Response: Thank you. We have undertaken an extensive corrections for both the language and the grammar.
In the present form the paper is not suitable for publication due to the following reasons:
- do not contains relevant and up-to-date information
- not focussed on real problems: milk EVs isolation and characterization, presence of milk fatty, casein micelles, etc.
- the text is weakly formulated, contains serious conceptual errors and the whole text is not established scientifically
- Tables are too general. I would prefer a summary of milk EV proteins in a Table, too
- The text seems to be rather a draft them a real scientific review paper
- Perhaps after a very strong amendment, the manuscript should be submitted again.
Response: We have now improved the overall quality of the manuscript by adding new text and tables. We trust that our revised manuscript covers the generalize concept of milk exosomes and their potential use for the management of cancer. Also, a separate list of protein table has been added as requested by the Reviewer 2.
Some minor comments:
- 2 is missing
Response: We are sorry for this mistake. We attached the figure to the submitted file.
- I advise to show at least some TEM pictures on milk EVs
Response: We have TEM images included but as we are using them for the original research article, it is not possible to include it in this review. For the sake of the learned reviewer’s comment, we have submitted it here in the response letter.

Round 2
Reviewer 2 Report
Review on revised manuscript entitled
The therapeutic potential of milk extracellular vesicles on colorectal cancer
I appreciate the authors’ effort to improve the manuscript (proteomics data, e.g.). However, there are still a lot of absences and shortcomings regarding the revised manuscript, hindering the publication.
1. There are some incomplete sentences, like ‘EVs has an immune role in the mid-1990s’.
2. Isolation protocols are not focussed on milk-derived EVs. There are general protocols for EV separation, but how these meet the requirements for isolation of pure milk EVs? I still do not found any mention on casein micelles, fat globules, etc., which could complicate traditional EV separation methods. As I know the literature, the most common milk EV isolation is the sucrose gradient centrifugation. Please study and add the relevant literature.
https://doi.org/10.3168/jds.2020-19849,
https://doi.org/10.3390/nu13082505
I insist my previous opinion:
“It would be nice to refer on the special case of milk EVs. My experience is that common EV isolation methods very often failed to produce vesicles from milk. Indeed, I would prefer to read about how others succeed with isolation, characterization. Again, characterization methods are very stereotyped and state-of-the art techniques like the concentration step by tangential flow filtration, microfluidic resistive pulse sensing, determination of protein or lipid concentrations are even not mentioned.”
3. Characterization method are still barely discussed. Table 2 is OK, however, some drawbacks are cutely written: ‘ Takes a lot of time and is very complicated’, “The NTA then analyzes the information.” „Dynamic light scattering (DLS) uses a laser beam transmitted over a solution of nanoparticles” –that is all about DLS.
I insist on my previous opinion:
“The reader expect some data about the general size of milk EVs. Description of TEM is again inacceptable. Even that a review article do not need to contain own new results, however, a scientific level is required. Are the researcher using cryo-TEM? How these milk EVs looks like? Can they be visualized without staining? What contrast-enhancing techniques are commonly used?”
- I advise to show at least some TEM pictures on milk EVs.
Regarding authors’ response: ‘We have intentionally used very simple language for explaining the methodological part of exosomal isolation and characterization. The main reason for this is to attract new researchers of the field to understand the cumbersome techniques in a simple way. This is written to invite the uninformed to the subject matter.’
I deeply condemn this response. Even a propagative work must be correct, clear and comprehensive with a scientific demand. Moreover, this is a scientific publication…
The revised manuscript is still not ready for publication. A thoroughly grammatical and stylistic check-up is needed, and the above comments were not addressed upon revision.
Author Response
I appreciate the authors’ effort to improve the manuscript (proteomics data, e.g.). However, there are still a lot of absences and shortcomings regarding the revised manuscript, hindering the publication.
1. There are some incomplete sentences, like ‘EVs has an immune role in the mid-1990s’.
R1. We thank the reviewer for this notion. We deleted this incomplete sentence.
2. Isolation protocols are not focussed on milk-derived EVs. There are general protocols for EV separation, but how these meet the requirements for isolation of pure milk EVs? I still do not found any mention on casein micelles, fat globules, etc., which could complicate traditional EV separation methods. As I know the literature, the most common milk EV isolation is the sucrose gradient centrifugation. Please study and add the relevant literature.
https://doi.org/10.3168/jds.2020-19849,
https://doi.org/10.3390/nu13082505
I insist my previous opinion:
“It would be nice to refer on the special case of milk EVs. My experience is that common EV isolation methods very often failed to produce vesicles from milk. Indeed, I would prefer to read about how others succeed with isolation, characterization. Again, characterization methods are very stereotyped and state-of-the art techniques like the concentration step by tangential flow filtration, microfluidic resistive pulse sensing, determination of protein or lipid concentrations are even not mentioned.”
R2. We thank the reviewer for this notion. We updated the text to mention these methods.
3. Characterization method are still barely discussed. Table 2 is OK, however, some drawbacks are cutely written: ‘ Takes a lot of time and is very complicated’, “The NTA then analyzes the information.” „Dynamic light scattering (DLS) uses a laser beam transmitted over a solution of nanoparticles” –that is all about DLS.
I insist on my previous opinion:
“The reader expect some data about the general size of milk EVs. Description of TEM is again inacceptable. Even that a review article do not need to contain own new results, however, a scientific level is required. Are the researcher using cryo-TEM? How these milk EVs looks like? Can they be visualized without staining? What contrast-enhancing techniques are commonly used?”
- I advise to show at least some TEM pictures on milk EVs.
R3. Response: We thank the reviewer for this comment. Yes, we tried Cryo-TEM but on another biological fluid and we provide the image. Sure, Cryo-TEM provides high contrast and preservation of the sample for a long time.
The protocol for Cryo-TEM was as following:
3.5 ul of the sample was applied to glow-discharged Quantifoil R1.2/1.3 Cu 300 grids (Quantifoil) and were flash frozen in liquid ethane using Vitrobot mark IV (Thermo Fisher Scientific) set at 100% humidity and 4°C for the preparation chamber and 5 s blot time. Cryo-EM micrographs were imaged on Glacios microcscope (Thermo Fisher Scientific) operated at an accelerating voltage of 200 kV with a 70 um C2 aperture at an indicated magnification of 73 kX. A Falcon III direct electron detector in linear mode was used to acquire images of the samples with a 100 um objective aperture.
Please see the attached file for the images.
4. Regarding authors’ response: ‘We have intentionally used very simple language for explaining the methodological part of exosomal isolation and characterization. The main reason for this is to attract new researchers of the field to understand the cumbersome techniques in a simple way. This is written to invite the uninformed to the subject matter.’
I deeply condemn this response. Even a propagative work must be correct, clear and comprehensive with a scientific demand. Moreover, this is a scientific publication…
R4. Response: We edited the text for better illustration.
5. The revised manuscript is still not ready for publication. A thoroughly grammatical and stylistic check-up is needed, and the above comments were not addressed upon revision.
R5. We thank the reviewer for the suggestion. We revised the manuscript through the MDPI English editing service with #45860 The text has been checked for correct use of grammar and common technical terms and edited to a level suitable for reporting research in a scholarly journal. MDPI uses experienced, native English-speaking editors.
